# Toward an Optimal Selection of Constraints for Terrestrial Reference Frame (TRF)

**Shize Song [1,2] , Zhongkai Zhang [3,4,5,*] and Guangli Wang [1,2]**

[1] Shanghai Astronomical Observatory, Chinese Academy of Sciences, No. 80 Nandan Road, Shanghai 200030, China; ssz@shao.ac.cn (S.S.); wgl@shao.ac.cn (G.W.)
[2] University of Chinese Academy of Sciences, 19A Yuquanlu, Beijing 100049, China
[3] College of Geospatial Information, Information Engineering University, Zhengzhou 450000, China
[4] Henan Industrial Technology Academy of Spatio-Temporal Big Data, Zhengzhou 450046, China
[5] The College of Geography and Environmental Science, Henan University, Kaifeng 475004, China
[*] Correspondence: zhang.astro@foxmail.com

**Abstract:** Given that the observations from current space geodetic techniques do not carry all the necessary datum information to realize a Terrestrial Reference System (TRS), and each of the four space geodetic techniques has limits, for instance: Very Long Baseline Interferometry (VLBI) ignores the center of mass and satellite techniques lack the TRS orientation, additional constraints have to be added to the observations. This paper reviews several commonly used constraints, including inner constraints, internal constraints, kinematic constraints, and minimum constraints. Moreover, according to their observation equations and normal equations, the similarities and differences between them are summarized. Finally, we discuss in detail the influence of internal constraints on the scale of VLBI long-term solutions. The results show that there is a strong correlation between the scale parameter and the translation parameter introduced by the combination model at the Institut National de l'Information Géographique et Forestière (IGN), and internal constraints force these two groups of parameters to meet certain conditions, which will lead to the coupling of scale and translation parameters and disturbing the scale information in VLBI observations. The minimum or kinematic constraints are therefore the optimum choices for TRF.

**Keywords:** TRF; constraints; correlation; optimal selection; combination model

## 1. Introduction

The International Terrestrial Reference System (ITRS) definition fulfills the following conditions [1]:

1. It is geocentric, and its origin is the center of mass for the whole Earth, including oceans and atmosphere;

2. The unit of length is the meter (Le Système International d'Unités (SI)). The scale is consistent with the geocentric coordinate time (TCG) time coordinate for a geocentric local frame, in agreement with the International Astronomical Union (IAU) and the International Union of Geodesy and Geophysics (IUGG) (1991) resolutions. (The mean rate of the coordinate time TCG coincides with the mean rate of the proper time of an observer situated at the geocenter (with the Earth removed), whereas the mean rate of the terrestrial time (TT) coincides with the mean rate of the proper time of an observer situated on the geoid. $TCG - TT \approx 0.7$ ppb [2].) This is obtained by appropriate relativistic modeling;

3. Its orientation was initially given by the Bureau International de l'Heure (BIH) orientation at 1984.0;

4. The time evolution of the orientation is ensured by using a no-net-rotation (NNR) condition with regards to horizontal tectonic motions over the whole Earth.

By defining the above datum definition, ITRS is implemented. The International Terrestrial Reference Frame (ITRF) is a long-term, linear reference frame, as defined by

the International Earth Rotation and Reference Systems Service (IERS) Conventions (2010). Thus, the ITRF datum information includes the origin, scale, orientation, and corresponding rates. As the ITRS realization, ITRF is the combination of the four space geodetic techniques (Global Navigation Satellite Systems (GNSS), Satellite Laser Ranging (SLR), VLBI, and Doppler Orbitography and Radio-positioning Integrated by Satellite (DORIS). As an unbiased-ranging technology, SLR uniquely determines the ITRF origin, which is related to the focal point of the satellite orbits [3,4]. Due to VLBI measuring extremely precise baseline lengths involving the speed of light [5] and the unambiguous nature of SLR measurements [6], the scale of the ITRF is provided by both VLBI and SLR. In order to ensure continuity, the orientation of ITRF is conventionally aligned with the BIH earth orientation parameter (EOP) series at 1984.0 [1,7].

However, observations from any space geodesy do not contain all the necessary datum information to completely define a TRS. For example, VLBI ignores the Earth center of mass, and satellite techniques including GNSS, SLR, and DORIS lack the orientation datum information. Therefore, for the realization of the ITRS, constraints have to be added to the observations to make the normal Equation (NEQ) invertible. The constraints are required to complete the rank deficiency of NEQ without causing the station network distortion and affecting the datum information of the observations, such as the scale of VLBI [8]. Several constraints commonly used in ITRF calculations include: inner constraints [9], minimum constraints [8], internal constraints [10], and kinematic constraints [9,11]. These constraints are reviewed in this work. Furthermore, according to their observation equations and normal equations, the similarities and differences between them are summarized.

In this paper, we only discuss constraints on the VLBI scale information in the VLBI intra-technique combination, i.e., stacking the time series. According to different intra-technique combination models, the corresponding datum constraint is selectable. IERS contains three ITRS combination centers (CCs): IGN, Deutsches Geodätisches Forschungsinstitut (DGFI-TUM) and Jet Propulsion Laboratory (JPL). The ITRFs provided by IGN and DGFI-TUM are secular reference frames, and the one from JPL is based on a Kalman filter approach producing time series of weekly solutions [12]. In this work, we assume that the ITRF model is long-term and linear within the scope of the IERS Conventions (2010) [1].

The computation strategy of DGFI-TUM is based on the combination at the NEQ level, while the ITRS CC at IGN is at the solution level. Before the combination, time series analysis of the input data is needed to improve the accuracy of the linear frame, which includes outlier detection, discontinuities, velocity changes, and estimation of seasonal signals. Outliers are detected and eliminated in the process of stacking time series and producing long-term solutions [13,14]. If the normalized residual exceeded a threshold of 3, the observations would be eliminated as outliers. After several iterations, all outliers are removed. The discontinuities and velocity variations in the time series of station positions are estimated by considering equipment changes from station log files [15] and earthquake information [16–18]. In this paper, we directly use the discontinuities and velocity change information provided by IGN (available at https://itrf.ign.fr/ITRF_solutions/2014/computation_strategy.php?page=2 (accessed on 22 January 2022)).For ITRF2014, seasonal signals of the stations were estimated during the stacking [13]. For DGFI-TUM's ITRS realization 2014 (DTRF2014), time series of atmospheric and hydrological non-tidal loading corrections provided by Tonie van Dam was used to reduce the influence of seasonal signals before the stacking (available at https://doi.pangaea.de/10.1594/PANGAEA.864046?format=html#download (accessed on 22 January 2022)). In this paper, seasonal signals are not estimated or corrected because it is not expected that the seasonal signals affect the ITRF datum information and the velocities of stations with more than 2.5 years of observations [13,19]. Before the stacking, postseismic deformation (PSD) for sites mainly caused by major earthquakes was modeled (corresponding data and subroutines available at https://itrf.ign.fr/ITRF_solutions/2014/ITRF2014_files.php (accessed on 22 January 2022)).

For the IGN combination, in order to provide the origin and orientation datum, minimum constraints are applied to the VLBI 24h sessions provided under the form of normal equations before stacking the time series. The addition of minimum constraints on the datum of orientation and origin to the VLBI free normal equations not only preserves its physical scale parameter but also allows its inversion [10]. The IGN intra-technique combination model is based on the seven-parameter similarity transformation [9,20]. In addition to calculating station coordinates and EOPs, the time series of the seven transformation parameters between each daily minimum constrained solution and the corresponding long-term stacked solution was also estimated at IGN. Since the transformation parameters were introduced to the accumulated NEQ by the combination model at IGN, two constraints could be selected in the intra-technique combination (the time series stacking), where the minimum constraints [8] were imposed over the station coordinates, and internal constraints [9,10] were applied over the time series of each of the seven transformation parameters. The DGFI model is based on the combination of the NEQs free from additional constraints. The VLBI Solution Independent Exchange (SINEX) files contain normal equation systems free from datum constraints and resulting from a combination of the Analysis Centers' (ACs) contributions at the NEQ level [21]. Therefore, DGFI-TUM directly stacks VLBI time series of NEQs to generate a multi-year normal equation system [14,22]. As the stacked normal equation does not contain the time series of the transformation parameters, DGFI-TUM can only impose conditions over station positions and velocities.

Since the constrained parameters are different (the internal constraints act on the time series of each of the seven transformation parameters, while the minimum constraints act on the station coordinates), it is necessary to compare the two constraints by means of specific calculations. Altamimi et al. compared the calculation results realized by internal and minimum constraints, respectively, [10]: (1) The post-fit residuals derived from the two methods are still the same; (2) the Helmert parameters between these two corresponding accumulated solutions (intra-technique combination) are different, which is estimated by the 14-parameter similarity transformation. By analyzing the observation equations and implicit conditions of these two constraints, the rationality of similar results (case 1) can be verified. However, the difference of the obtained transformation parameters (case 2) indicates that the datum of the long-term solutions corresponding to the two constraints is inconsistent. By calculating the scales at epoch 2010.0 and scale rates of DTRF2014 DORIS, GNSS, SLR, and VLBI with respect to ITRF2014 [23], there is a non-negligible linear trend between the long-term solutions of the same technique. Although the ITRF2014 scale is defined by the average of the VLBI and SLR scale [13], the scale of the multi-technique combination solutions is determined by the long-term solution (intra-technique combination). Therefore, the scale between the same technologies of DTRF2014 and ITRF2014 should be a constant offset rather than a linear trend. By calculating scale offsets and their rates calculated from the 14-parameter Helmert transformation of the VLBI and SLR single-technique solutions of (intra-technique combination) DGFI-TUM and IGN (transformation epochs are at 2000.0 and 2010.0) [24], the results show that there is a linear trend between the scales of single-technique solutions (the intra-technique combination) of IGN and DGFI-TUM. Considering the nearly 40 years worth of observation history for VLBI and SLR, this linear trend is not negligible. According to the preliminary calculations of ITRF2020, the scale difference between SLR and VLBI is about 3 mm [25], versus 8.7 mm in ITRF2014 [13]. However, IGN has not provided the modified combination method. In DTRF2014, which used minimum constraints instead of internal constraints, the scales of SLR and VLBI are considered to be statistically equal ($\pm 3.3$ mm) [22]. In algebraic terms, minimum constraints can complete the NEQ rank deficiency of long-term solutions and not more [8]. Moreover, when we check the VLBI minimum constraint solution (input solutions to ITRF2014 [21] and the long-term solutions produced in this work), the origin and orientation are indeed expressed in an a priori reference frame. Therefore, the minimum constraints do not affect the scale information of VLBI observations. Based on the above analysis, we believe that internal constraints may affect the datum of long-term solutions obtained from the intra-

technical combination. This paper will describe how internal constraints affect the scale datum of VLBI technology, considering the intra-technique combination model of IGN and DGFI-TUM, respectively.

The main objective of this article is to review and compare several commonly used constraints and select the optimal ones for TRF.

Section 2 introduces the combination model of IGN and DGFI-TUM, reviews kinematic constraints, minimum constraints, internal and inner constraints, and summarizes the relationship between these constraints. Section 3 gives the results of four VLBI long-term solutions with different constraints and intra-technique combination models. Section 4 discusses how internal constraints affect the VLBI long-term solution. Section 5 concludes.

## 2. Materials and Methods

The scale difference between the VLBI long-term solutions of IGN and DGFI-TUM [24] may occur for two reasons: the combination model and constraints. This section briefly introduces the combination models of IGN and DGFI-TUM. The ITRF combination procedure can be divided into two main steps: the intra-technique and inter-technique combination [13,14]. We review several commonly used constraints, including inner constraints, internal constraints, kinematic constraints, and minimum constraints. Moreover, according to their observation equations and normal equations, the similarities and differences between them are summarized. The research on constraints is quite well-established. Based on existing results (see Section 1), this section summarizes the relationship between these constraints. For this purpose, we review these constraints and study the relationship between them by comparing their preconditions, observation equations, and normal equations. In order to avoid the interference of other factors on the VLBI long-term solutions, the same time series analysis method was used.

### 2.1. Combination Model at IGN

Since the computation strategy of IGN is based on the combination of solutions, all input data are converted to the minimum constraint solutions before the combination. The combination model of IGN can be summarized as [20]:

$$
\begin{cases}
\boldsymbol{X}_s^i = \boldsymbol{X}_c^i + (t_s^i - t_0)\dot{\boldsymbol{X}}_c^i + \boldsymbol{T}_k + D_k \boldsymbol{X}_c^i + \boldsymbol{R}_k \boldsymbol{X}_c^i \\
\quad + (t_s^i - t_k)[\dot{\boldsymbol{T}}_k + \dot{D}_k \boldsymbol{X}_c^i + \dot{\boldsymbol{R}}_k \boldsymbol{X}_c^i] \\
\dot{\boldsymbol{X}}_s^i = \dot{\boldsymbol{X}}_c^i + \dot{\boldsymbol{T}}_k + \dot{D}_k \boldsymbol{X}_c^i + \dot{\boldsymbol{R}}_k \boldsymbol{X}_c^i
\end{cases}
\tag{1}
$$

$$
\begin{cases}
x_s^p &= x_c^p + R_{yk} \\
y_s^p &= y_c^p + R_{xk} \\
UT_s &= UT_c - \dfrac{1}{f} R_{zk} \\
\dot{x}_s^p &= \dot{x}_c^p \\
\dot{y}_s^p &= \dot{y}_c^p \\
LOD_s &= LOD_c
\end{cases}
\tag{2}
$$

where positions (at epoch $t_s^i$) and velocities of each station $i$ of a single-technique solution ($s$ = VLBI, SLR, GNSS or DORIS) are represented by $X_s^i$ and $\dot{X}_s^i$, respectively, and those of the combined solution $c$ by $X_c^i$ at reference epoch $t_0$ and $\dot{X}_c^i$. For each solution $s$ expressed in the frame $k$ at epoch $t_k$, $\boldsymbol{T}_k$ is the translation vector including three origin components ($T_x$, $T_y$, $T_z$), $\boldsymbol{R}_k$ the rotation matrix composed of 3 rotation parameters ($R_x$, $R_y$, and $R_z$), and $D_k$ the scale factor. The dotted parameters are their derivatives with respect to time. The EOPs contain pole coordinates ($x_s^p$, $y_s^p$) and universal time ($UT_s$), and $\dot{x}_s^p$, $\dot{y}_s^p$, and $LOD_s$ are their rates. The conversion factor $f$ from universal time into sidereal time is equal to 1.002737909350795.

Since the daily or weekly solution $X_s^i$ does not involve station velocity $\dot{X}_s^i$ and transformation parameter rates, all of Equation (2) and only the first line of Equation (1) are used in the intra-technique combination. In the second step of the inter-technique combination, the two equations above the four long-term solutions are combined.

### 2.2. Combination Model at DGFI-TUM

The DGFI-TUM model is based on the combination of the NEQs free from additional constraints. Its combination model can be described by Equations (3)–(7) [14].

$$\begin{cases} \hat{x} = N^{-1}y \\ \hat{\sigma}^2 = \dfrac{l^T Pl - y^T \hat{x}}{n - u} \end{cases} \tag{3}$$

where $n$ and $u$ represent the number of observations and estimates, respectively, $N$ the matrix of NEQ, $\hat{x}$ the estimates, $\hat{\sigma}^2$ the posteriori variance factor, $y = A^T Pl$ the product, $l^T Pl$ the square sum of the observed vector minus the computed vector (O-C). $A$, $l$, and $P$ are the coefficient matrix, the observation vector, and the weight matrix of the observations, respectively. The covariance matrix $C_{\hat{x}\hat{x}}$ of the estimates is described by Equation (4).

$$C_{\hat{x}\hat{x}} = \hat{\sigma}^2 N^{-1} \tag{4}$$

The final NEQ (Equation (3)) is obtained in two steps. In the intra-technique combination, time series of NEQs from the Technique Centres (TCs) are stacked to one accumulated NEQ $\hat{x}_i = N_i^{-1} y_i$ ($i$ is one of the four space geodetic technologies). In the second step, the accumulated NEQs (obtained from step 1) of the different techniques are combined:

$$N = \sum_i \lambda_i N_i \tag{5}$$

$$y = \sum_i \lambda_i y_i \tag{6}$$

$$l^T Pl = \sum_i \lambda_i (l^T Pl)_i \tag{7}$$

where $\lambda_i$ represents the weight factors estimated for the techniques.

### 2.3. Kinematic Constraints

Kinematic constraints have physical meanings [9]: (a) with respect to the origin by imposing constant reference coordinates for the barycenter of the station network, (b) with respect to orientation by imposing zero relative angular momentum for the network stations with the assumption that mass points (stations) have equal masses, and (c) with respect to the scale by imposing a constant mean quadratic size (related to the distances from stations to their barycenter). The observation equations of NNR and no-net-translation (NNT) conditions are given below. For kinematic constraints related to scale, please refer to [9,11].

(1) NNT

$$\sum_{i=1}^{N} \delta x_i = 0 \tag{8}$$

$$\sum_{i=1}^{N} \delta v_i = 0 \tag{9}$$

(2) NNR

$$\sum_{i=1}^{N} [x_i^{ap} \times] \delta x_i = 0 \tag{10}$$

$$\sum_{i=1}^{N}[x_i^{ap}\times]\delta v_i = 0 \tag{11}$$

$$[x_i^{ap}\times] = \begin{bmatrix} 0 & z_i^0 & -y_i^0 \\ -z_i^0 & 0 & x_i^0 \\ y_i^0 & -x_i^0 & 0 \end{bmatrix} \tag{12}$$

where $\delta x_i$ is the correction of the station position $x_i$, and its prior value is $x_i^{ap}$. $\delta v_i$ is the correction of the station velocity $v_i$, and its prior value is $v_i^{ap}$.

For simplicity, we only give NEQs of NNT and NNR with respect to the station position (corresponding to Equations (8) and (10)).

The matrix form of NNT is:

$$A_{NNT} \cdot \delta x = \underbrace{\begin{bmatrix} 1 & 0 & 0 & \cdots & 1 & 0 & 0 & \cdots & 1 & 0 & 0 \\ 0 & 1 & 0 & \cdots & 0 & 1 & 0 & \cdots & 0 & 1 & 0 \\ 0 & 0 & 1 & \cdots & 0 & 0 & 1 & \cdots & 0 & 0 & 1 \end{bmatrix}}_{3\times 3N} \underbrace{\begin{bmatrix} \delta x_1 \\ \delta y_1 \\ \delta z_1 \\ \vdots \\ \delta x_i \\ \delta y_i \\ \delta z_i \\ \vdots \\ \delta x_N \\ \delta y_N \\ \delta z_N \end{bmatrix}}_{3N\times 1} = \underset{3\times 1}{\mathbf{0}} \tag{13}$$

where $N$ is the number of stations.

The NEQ of NNT is:

$$\underset{3N\times 3N}{N_{NNT}}\delta x = (A_{NNT}^T P_x A_{NNT}) \cdot \delta x = 0 \tag{14}$$

where the diagonal matrix $P_x$ is the weight matrix. $P_x$ is equivalent to a coefficient $p_x$ in Equation (14). The coefficient matrix $N_{NNT}$ of NEQ is:

$$N_{NNT} = p_x \begin{bmatrix} 1 & 0 & 0 & \cdots & 1 & 0 & 0 & \cdots & 1 & 0 & 0 \\ 0 & 1 & 0 & \cdots & 0 & 1 & 0 & \cdots & 0 & 1 & 0 \\ 0 & 0 & 1 & \cdots & 0 & 0 & 1 & \cdots & 0 & 0 & 1 \\ \vdots & \vdots & \vdots & \ddots & \vdots & \vdots & \vdots & \ddots & \vdots & \vdots & \vdots \\ 1 & 0 & 0 & \cdots & 1 & 0 & 0 & \cdots & 1 & 0 & 0 \\ 0 & 1 & 0 & \cdots & 0 & 1 & 0 & \cdots & 0 & 1 & 0 \\ 0 & 0 & 1 & \cdots & 0 & 0 & 1 & \cdots & 0 & 0 & 1 \end{bmatrix}$$

The matrix form of NNR is:

$$
A_{NNR} \cdot \delta x = \begin{bmatrix} [x_1^{ap} \times] & \cdots & [x_i^{ap} \times] & \cdots & [x_N^{ap} \times] \end{bmatrix}_{3 \times 3N} \begin{bmatrix} \delta x_1 \\ \delta y_1 \\ \delta z_1 \\ \vdots \\ \delta x_i \\ \delta y_i \\ \delta z_i \\ \vdots \\ \delta x_N \\ \delta y_N \\ \delta z_N \end{bmatrix}_{3N \times 1} = \underset{3 \times 1}{\mathbf{0}}
\tag{15}
$$

The NEQ of NNR is:

$$
\underset{3N \times 3N}{N_{NNR}} \delta x = (A_{NNR}^T P_x A_{NNR}) \cdot \delta x = 0
\tag{16}
$$

The coefficient matrix $N_{NNR}$ of NEQ is:

$$
N_{NNR} = p_x \begin{bmatrix} [x_1^{ap} \times]^T [x_1^{ap} \times] & \cdots & [x_i^{ap} \times]^T [x_i^{ap} \times] & \cdots & [x_N^{ap} \times]^T [x_N^{ap} \times] \end{bmatrix}
$$

### 2.4. Minimum Constrains

We review the minimum constraints [8] and give the corresponding normal equations. Before giving the minimum constraint, the similarity transformation of 14 Helmert parameters is introduced. Assuming that any solution $X_s$ only includes the station position and velocity, then $X_s$ can be transformed into the ITRF solution $X$ by the 14 Helmert parameters, and the linearized matrix form is directly given, as follows:

$$
X = X_s + A\theta
\tag{17}
$$

where $\theta = (T1, T2, T3, D, R1, R2, R3, \dot{T}1, \dot{T}2, \dot{T}3, \dot{D}, \dot{R}1, \dot{R}2, \dot{R}3)$ are 14 Helmert parameters, and $T1, T2$, and $T3$; $D$; $R1, R2$, and $R3$ represent three translation parameters; one scale parameter; and three rotation parameters, respectively, and the partial derivative symbol represents their corresponding rates. A is the design matrix derived from 14 to parameter similarity transformation, where $N$ is the number of stations in the solution $X_s$, and $(\ldots, x_i^0, \ldots)$ represnts approximate positions of the station $i$:

$$
A = \begin{bmatrix}
\cdot & \cdot & \cdot & \cdot & \cdot & \cdot & \cdot & \cdot & \cdot & \cdot & \cdot & \cdot & \cdot & \cdot \\
1 & 0 & 0 & x_i^0 & 0 & z_i^0 & -y_i^0 & & & & & & & \\
0 & 1 & 0 & y_i^0 & -z_i^0 & 0 & x_i^0 & & & \approx 0 & & & & \\
0 & 0 & 1 & z_i^0 & y_i^0 & -x_i^0 & 0 & & & & & & & \\
& & & & & & & 1 & 0 & 0 & x_i^0 & 0 & z_i^0 & -y_i^0 \\
& & \approx 0 & & & & & 0 & 1 & 0 & y_i^0 & -z_i^0 & 0 & x_i^0 \\
& & & & & & & 0 & 0 & 1 & z_i^0 & y_i^0 & -x_i^0 & 0 \\
\cdot & \cdot & \cdot & \cdot & \cdot & \cdot & \cdot & \cdot & \cdot & \cdot & \cdot & \cdot & \cdot & \cdot
\end{bmatrix}
\tag{18}
$$

Using an unweighted least-squares adjustment, $\theta$ can be solved:

$$
\theta = (A^T A)^{-1} A^T (X - X_S)
\tag{19}
$$

The $\theta$ in Equation (19) is regarded as the virtual observation value, and the error equation is given:

$$
v_\theta = B(X - X_s)
\tag{20}
$$

where $B = (A^T A)^{-1} A^T$, and $v_\theta$ is the residual of $\theta$. Because of the regularity of the matrix $A^T A$, it is even possible to use $B = A^T$ [26]. In geodetic analysis, the TRF difference between $X$ and $X_s$ is very small, so a prior value of the virtual observation value $\theta$ can be set to zero.

The NEQ of the minimum constraints derived from Equation (20) is:

$$(A\Sigma_\theta^{-1}A^T)(X - X_s) = 0 \tag{21}$$

where a diagonal matrix composed of small empirical variances corresponding to the 14 Helmert parameters is represented by $\Sigma_\theta$.

For comparison with kinematic constraints, Equation (21) is simplified to:

$$p_x AA^T \delta x = 0 \tag{22}$$

where the diagonal matrix $\Sigma_\theta^{-1}$ can be simplified as a coefficient $p_x$.

### 2.5. Equivalence between Minimum Constraints and Kinematic Constraints

When only considering the origin datum (i.e., columns 1, 2, and 3 of the matrix $A$ in Equation (18)), its normal equation of minimum constraints (see Equation (23)) is equivalent to that of kinematic constraints (see Equation (14)):

$$p_x AA^T \delta x = p_x \begin{bmatrix} 1 & 0 & 0 & \cdots & 1 & 0 & 0 & \cdots & 1 & 0 & 0 \\ 0 & 1 & 0 & \cdots & 0 & 1 & 0 & \cdots & 0 & 1 & 0 \\ 0 & 0 & 1 & \cdots & 0 & 0 & 1 & \cdots & 0 & 0 & 1 \\ \vdots & \vdots & \vdots & \ddots & \vdots & \vdots & \vdots & \ddots & \vdots & \vdots & \vdots \\ 1 & 0 & 0 & \cdots & 1 & 0 & 0 & \cdots & 1 & 0 & 0 \\ 0 & 1 & 0 & \cdots & 0 & 1 & 0 & \cdots & 0 & 1 & 0 \\ 0 & 0 & 1 & \cdots & 0 & 0 & 1 & \cdots & 0 & 0 & 1 \end{bmatrix} = 0 \tag{23}$$

Similarly, the orientation constraints implemented by these two constraints are also equivalent, only considering columns 5, 6, and 7 of the matrix $A$. We checked the VLBI minimum constraint solution (input solutions to ITRF2014 [21] and the long-term solutions produced in this work), with the results showing that the origin and orientation datum defined by the minimum constraint conform to the kinematic constraints.

The NEQs of the datum rates constraints between the minimum constraints and the kinematic constraints are also equivalent.

### 2.6. Internal Constraints

In the intra-technique combination at IGN, internal constraints are used to define the datum of ITRF (scale and origin) [10,13,20]. The intra-technique combination model (see Equation (1)) produces time series of 7 Helmert transformation parameters between the input and a single-technique combination solution. With the assumption that the linear time evolution for the station positions and transformation parameters, we can write for each of the 7 transformation parameters $P_k$ (at epoch $t_k$) [10]:

$$P_k = P_k(t_0) + (t_k - t_0)\dot{P}_k \tag{24}$$

where $t_0$ represents the selected reference epoch of the combination solution.

Then, the least-squares adjustment can yield the following normal equation system of Equation (24):

$$\begin{pmatrix} K & \sum_{k \in K}(t_k - t_0) \\ \sum_{k \in K}(t_k - t_0) & \sum_{k \in K}(t_k - t_0)^2 \end{pmatrix} \begin{pmatrix} P_k(t_0) \\ \dot{P}_k \end{pmatrix} = \begin{pmatrix} \sum_{k \in K} P_k \\ \sum_{k \in K}(t_k - t_0)P_k \end{pmatrix} \tag{25}$$

The minimal/intrinsic constraints are used to impose some conditions on these transformation parameters:

$$\begin{cases} P_k(t_0) & = 0 \\ \dot{P}_k & = 0 \end{cases} \tag{26}$$

Imposing the two conditions implied in Equation (26) to NEQ (25), the right term of NEQ must be zero to ensure a unique zero solution. Internal constraints are derived:

$$\begin{cases} \sum\limits_{k \in K} P_k & = 0 \\ \sum\limits_{k \in K} \dfrac{P_k}{(t_k - t_0)^{-1}} & = 0 \end{cases} \tag{27}$$

### 2.7. Inner Constraints

Altamimi and Dermanis discuss and derive inner constraints and kinematic constraints [9] and conclude that the partial inner constraints for station parameters may coincide with the kinematical constraints, while partial inner constraints related to transformation parameters are equivalent to internal constraints. Inner constraints are given directly (see literature for detailed derivation):

$$\sum_{i=1}^{N} \delta \boldsymbol{x}_{0i} - \sum_{k=1}^{M} \boldsymbol{d}_k = 0 \tag{28}$$

$$\sum_{i=1}^{N} \delta \boldsymbol{v}_i - \sum_{k=1}^{M} (t_k - t_0) \boldsymbol{d}_k = 0 \tag{29}$$

$$\sum_{i=1}^{N} [\boldsymbol{x}_{0i}^{ap} \times] \delta \boldsymbol{x}_{0i} + \sum_{k=1}^{M} \boldsymbol{\theta}_k = 0 \tag{30}$$

$$\sum_{i=1}^{N} [\boldsymbol{x}_{0i}^{ap} \times] \delta \boldsymbol{v}_i + \sum_{k=1}^{M} (t_k - t_0) \boldsymbol{\theta}_k = 0 \tag{31}$$

$$\sum_{i=1}^{N} (\boldsymbol{x}_{0i}^{ap})^T \delta \boldsymbol{x}_{0i} - \sum_{k=1}^{M} s_k = 0 \tag{32}$$

$$\sum_{i=1}^{N} (\boldsymbol{x}_{0i}^{ap})^T \delta \boldsymbol{v}_i - \sum_{k=1}^{M} (t_k - t_0) s_k = 0 \tag{33}$$

where $\boldsymbol{d}_k$, $\boldsymbol{\theta}_k$, and $s_k$ represent translation, rotation, and scale transformation parameters, respectively.

The partial inner constraints for station parameters (the left parts of Equations (28)–(33)) are the exact kinematical constraints (see Section 2.3), while the partial inner constraints for transformation parameters are equivalent to internal constraints (see Section 2.6).

### 2.8. Relationship between Constraints

According to Sections 2.3–2.5, we can conclude that the minimum constraints and kinematic constraints are equivalent. Inner constraints unify internal constraints and kinematic constraints (see Section 2.3, 2.6 and 2.7).

Compared with the minimum constraints imposed over station parameters, the internal constraints act on the time series of Helmert parameters between each daily or weekly frame *k* and the intra-technique combination solution (see the first line of Equation (1), i.e., intra-technique combination). The internal constraint contains a condition that the conversion parameter is 0 (see Equation (26)), while the a priori value of $\boldsymbol{\theta}$, the virtual observation value of the minimum constraints, is also 0 (see Section 2.3). Variances of the transformation parameters $\boldsymbol{\theta}$ are small in minimum constraints, while the transformation parameters in internal constraints have no statistical significance (or can be seen as infinites-

imal). Therefore, the minimum constraints and internal constraints have both similarities and differences. The long-term solutions realized by these two constraints are compared at IGN, with the results showing that the post-fit residuals of the two methods are still the same and the transformation parameters between the two corresponding long-term accumulated solutions (intra-technique combination) are different, which is estimated by the 14-parameter similarity transformation.

However, only comparing the forms and results between the minimum constraints and internal constraints cannot explain the reasons for these differences. A detailed analysis of the process of intra-technique combination helps us understand how the internal constraints affect the datum of long-term solutions.

## 3. Results

Firstly, according to the VLBI intra-technique combination, we compare the scale of the long-term solutions realized by the internal constraints and the minimum constraints. Three years (2004.1.5–2006.12.29) of the time series of 24 h session data are available, which are provided by the International VLBI Service for Geodesy and Astrometry (IVS) for the ITRF2014 in the SINEX format [21]. Except for estimating seasonal signals of station position, the input data were analyzed using the same time series analysis method as ITRF2014 to eliminate the effects of discontinuity, velocity variation, PSD, and outliers [13]. We performed four stacking tests. Test 1, 2, and 3 use the IGN combination model (see Section 2.1), while test 4 uses the DGFI-TUM combination model (see Section 2.2). Before the stacking, minimum constraints are applied to inputs (NEQs) of tests 1, 2, and 3 to determine the origin and orientation datum. We extend the constrained NEQs of tests 1, 2, and 3 by 6 or 7 parameters of a similarity transformation (see Table 1).

**Table 1.** Extending NEQs by transformation parameters.

| Parameters | Test 1 | Test 2 | Test 3 | Test 4 |
|---|---|---|---|---|
| Translation | Yes | Yes | Yes | No |
| Rotation | Yes | Yes | Yes | No |
| Scale | Yes | No | Yes | No |

Tests 1 and 3 are extended by seven transformation parameters. Test 2 is extended by six parameters, corresponding to translation and rotation.

After extending the obtained NEQs by the station velocity parameter, the time series of NEQs are combined to one normal equation system. Minimum constraints or internal ones are applied to the accumulated normal equations of the four tests to complete the rank deficiency. For test 1, we choose internal constraints for the translation and scale components and the minimum constraint approach to define the orientation datum. For test 2, since scale parameters are not extended, we only choose minimum constraints for the translation and rotation components. For test 3, we choose minimum constraints for the translation and rotation components and the internal constraint approach to define the scale datum. For test 4, we choose minimum constraints for the translation and rotation components.

The scale parameters of four long-term solutions with respect to the ITRF2014 VLBI solution are shown in Table 2.

**Table 2.** The scale parameters of four long-term solutions with respect to the ITRF2014 VLBI solution.

| Epoch | Parameter | Test 1 | Test 2 | Test 3 | Test 4 |
|---|---|---|---|---|---|
| 2004.1.5 | | 0.4268 | 0.2777 | 0.3485 | 0.2757 |
| 2005.1.1 | Scale (ppb) | 0.4203 | 0.4347 | 0.4612 | 0.4357 |
| 2006.12.29 | | 0.4072 | 0.7500 | 0.6878 | 0.7570 |

Since we discuss linear frames in this work, the scale parameters at two epochs can reflect the scale rate.

The scale parameters of tests 2 and 4 are equal (see Table 2), and the difference between their station coordinates is negligible, indicating that the combination model does not affect the scale datum of VLBI. Compared with test 1, there is a linear trend (0.16 ppb/yr) on the scale of test 4. By estimating a 14-parameter similarity transformation, Altamimi et al. compared DTRF2014 solutions to the ITRF2014, with the result showing that there is a linear trend term between the VLBI solutions of DTRF2014 and ITRF2014 (see Figure 1) [23].

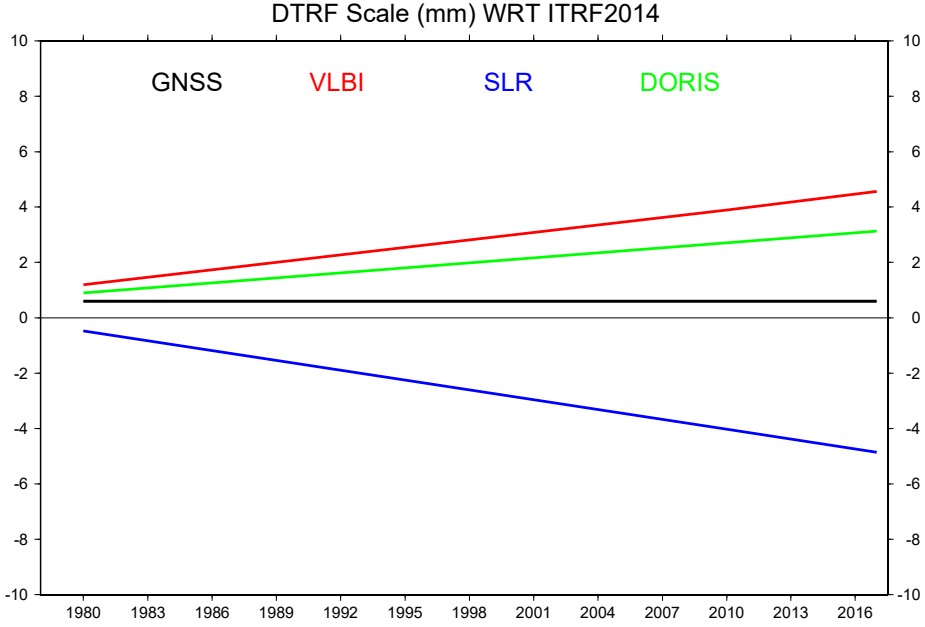

**Figure 1.** DTRF2014 scale with respect to ITRF2014.

Angermann et al. estimated the difference of VLBI single-technique solutions (intra-technique combination) of DGFI-TUM and IGN by 14 Helmert parameters, and the results showed that there is a linear trend between scales of these two solutions (see Figure 2) [24]. In order to minimize the scale impact for these two techniques, the scale of the ITRF2014 is defined in such a way that there is a zero scale factor (at epoch 2010.0) and a zero scale rate with respect to the average of the explicit scales and scale rates of the VLBI and SLR solutions [13,20]. Therefore, there is a constant offset in Figure 1 with respect to Figure 2.

For tests 1 and 3, the combination model of the two tests is the same, and the scale datum is realized by internal constraints. However, the scales of the two tests are inconsistent. Therefore, we have reason to suspect that there is a correlation between the time series of transformation parameters (especially scale and translation) introduced by the intra-technique combination. Moreover, in order to satisfy the internal constraint conditions, the scale and translation parameters may be mutually absorbed. In other words, the linear trend in VLBI scales is caused by internal constraints.

Although the VLBI inputs submitted to ITRF2014 are unconstrained normal equations, tests 1 and 3 are combined on the solution level, that is, all NEQs are applied with minimum constraints before superposition. Therefore, it is necessary to check the singularity of NEQs after setting up seven transformation parameters and station velocity to avoid the influence of over-constraints on the datum. Without loss of generality, the EOP and Helmert parameters are eliminated from the NEQs, and only the station coordinate parameters (including position and velocity) are retained. $n_1, n_2, \ldots, n_m$ ($m$ is the number of station coordinate parameters) are the row or column vectors of NEQs. From a geometric point of view, the lack of the NEQ datum information leading to the rank deficiency of the NEQ is based on the fact that the columns of the normal matrix are orthogonal to the corresponding columns of the transformation matrix (see Equation (18)) [27]. The column vectors of the

14-Helmert transformation matrix are expressed as $g_1, g_2, \ldots, g_{14}$. The rank deficiency of NEQs can be calculated by Equation (34):

$$< n, g > = \begin{bmatrix} \dfrac{n_1^T g_1}{\|n_1\| \cdot \|g_1\|} & \dfrac{n_1^T g_2}{\|n_1\| \cdot \|g_2\|} & \cdots & \dfrac{n_1^T g_{14}}{\|n_1\| \cdot \|g_{14}\|} \\ \dfrac{n_2^T g_1}{\|n_2\| \cdot \|g_1\|} & \dfrac{n_2^T g_2}{\|n_2\| \cdot \|g_2\|} & \cdots & \dfrac{n_2^T g_{14}}{\|n_2\| \cdot \|g_{14}\|} \\ \vdots & \vdots & \cdots & \vdots \\ \dfrac{n_m^T g_1}{\|n_m\| \cdot \|g_1\|} & \dfrac{n_m^T g_2}{\|n_m\| \cdot \|g_2\|} & \cdots & \dfrac{n_m^T g_{14}}{\|n_m\| \cdot \|g_{14}\|} \end{bmatrix} \tag{34}$$

where $< n, g >$ is the cosine value of the angle between the two vectors.

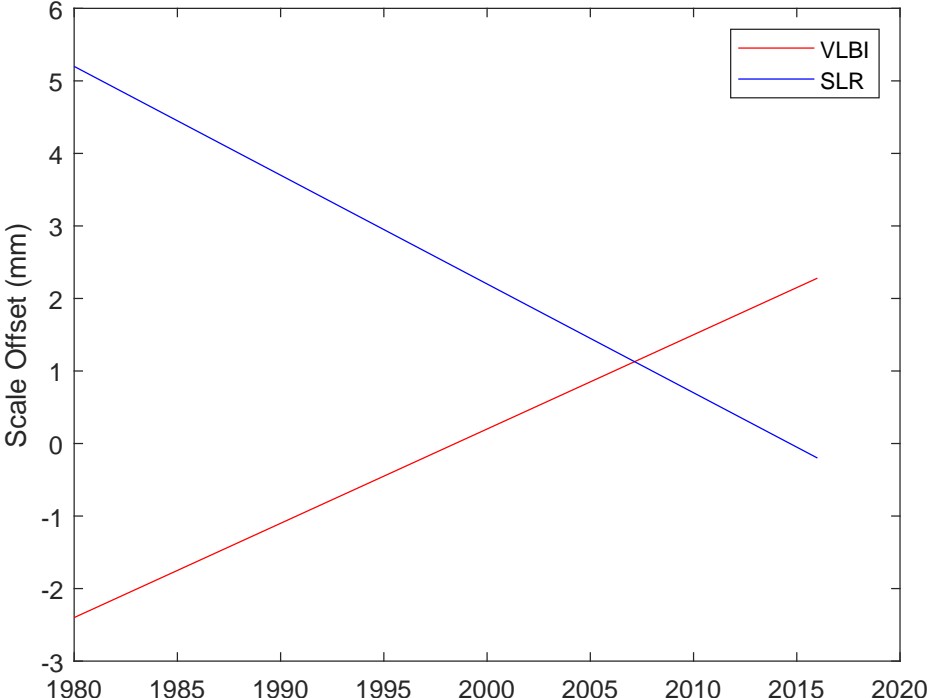

**Figure 2.** The linear trend between scales of the VLBI (red) and SLR (blue) long-term solutions from DGFI-TUM and IGN.

The singularity of NEQs can also be analyzed by calculating the orthogonality between the eigenvector of the coefficient matrix of NEQs and the similarity transformation matrix [27]. According to the above two singular analyses of NEQs, we can conclude that the long-term solutions obtained in this paper are not over-constrained. In other words, the scale datum of tests 1 and 3 is implemented only by internal constraints.

## 4. Discussion

In this section, we will check the correlation between the Helmert parameters of the session-wise NEQs. Firstly, the minimum constrained solutions of NEQs are substituted into the seven-parameter similarity transformation model. Then, the Pearson correlation coefficient between the transformation parameters in each normal equation of the similarity transformation is calculated. Figure 3 shows a strong correlation between scale and translation parameters and a weak correlation between scale and rotation parameters.

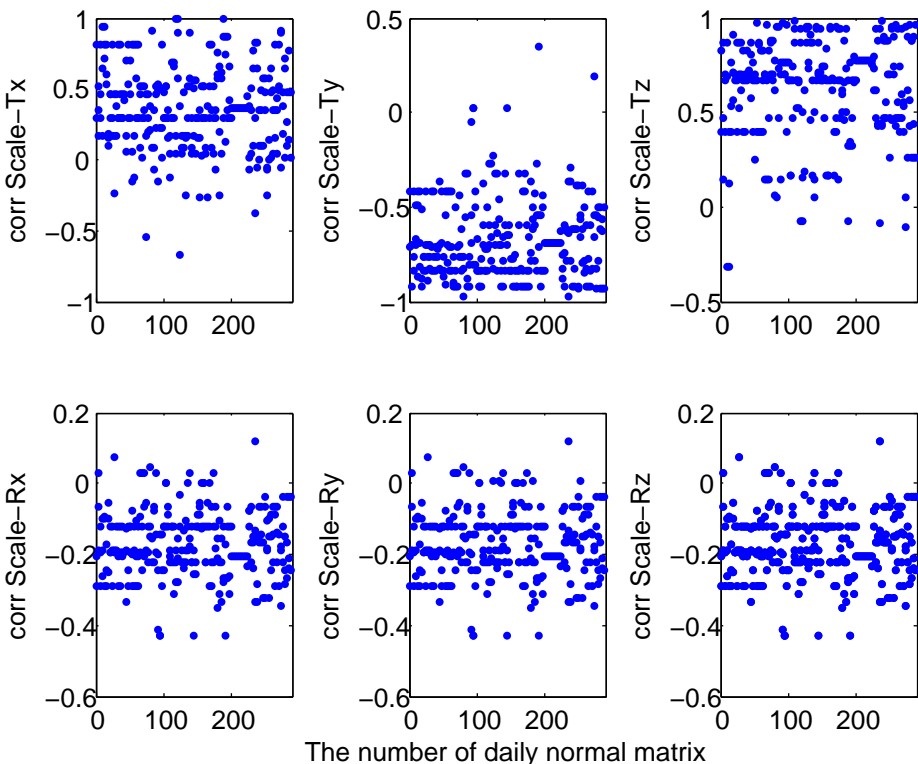

**Figure 3.** Correlation between scale parameters and other Helmert parameters.

Figure 4 shows the time series of translation parameters with respect to the long-term solutions of tests 1 (blue points on the left) and 2 (red points on the right).

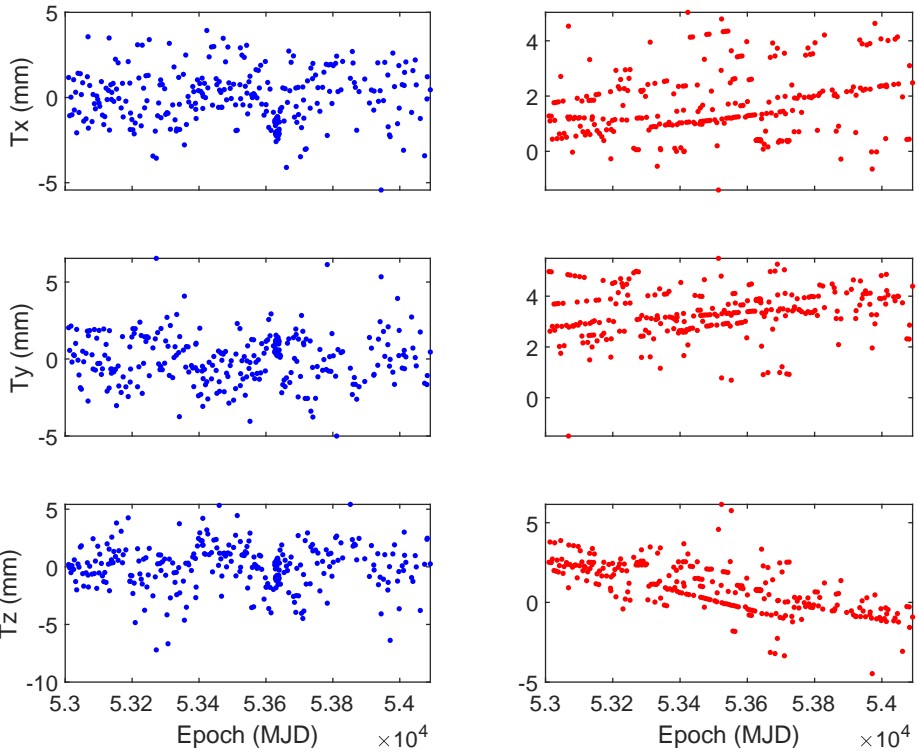

**Figure 4.** Time series of translation parameters with respect to the long-term solutions of tests 1 (blue) and 2 (red). MJD is Modified Julian Day.

These two results above confirm our previous assumption that there is a strong correlation between scale and translation parameters, and the internal constraint makes scale and translation parameters absorb each other, thus interfering with the scale of VLBI.

Figure 5 shows a weak correlation between translation and rotation parameters. The difference between the rotation parameters of tests 1 and 2 is at the *μas* level, that is, the orientation of the two frames can be considered the same.

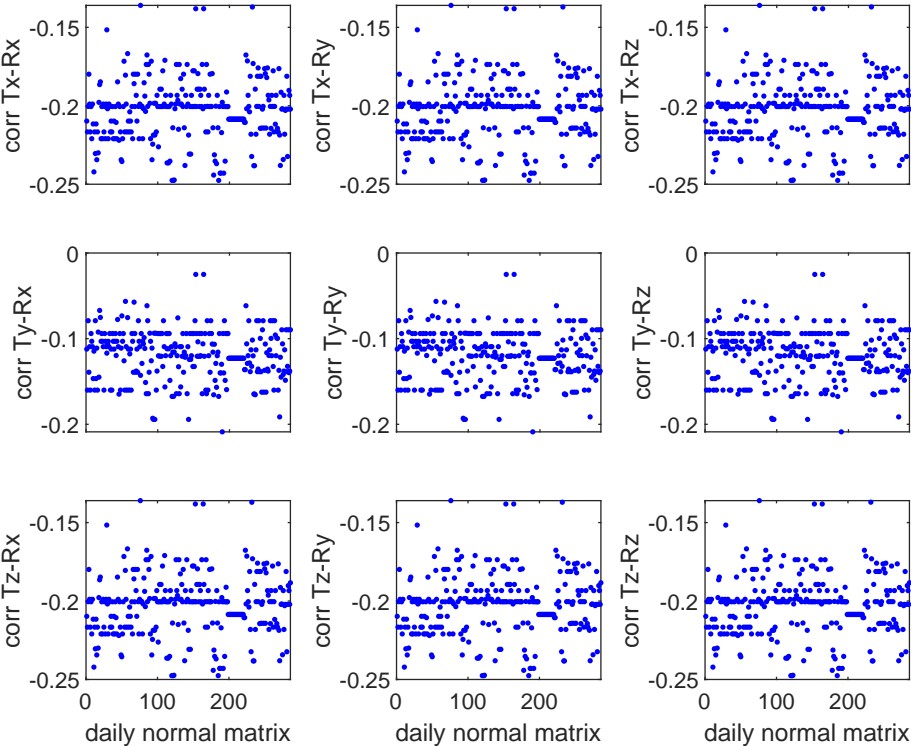

**Figure 5.** Correlation between translation and rotation parameters.

It should be noted that the internal constraints have strict requirements on stations' coverage of IVS sessions. The poor station coverage of IVS sessions leads to a strong correlation between rotation and other transformation parameters. ITRF2014 has to exclude the IVS sessions without more than four stations and those with poor coverage [13], although the excluded data are not outliers. Figures 6 and 7 show the station distribution of the two excluded sessions, corresponding to a session of no more than four stations and one with regional coverage, respectively.

Since the minimum constraints act on the station coordinates, the effect of these excluded sessions on tests 2 and 4 is not significant. Therefore, DGFI-TUM did not exclude VLBI data [28].

A comparison between solutions of the intra-technique combination [23] (see Figure 1) or inter-technique combination [24] (see Figure 2) shows that the scale of VLBI has a significant linear trend with respect to SLR in ITRF2014. With several tests showing that the scale offset between SLR and VLBI is less than 3.3 mm at 2000.0 [29], the scales of SLR and VLBI in the DTRF2014 was assumed to be statistically equal [22]. According to the preliminary calculations in the ITRF2020, the scale difference between SLR and VLBI is about 3 mm [25], versus 8.7 mm in ITRF2014 [13]. ITRF2020 does not provide a corresponding combination method. However, according to our calculations, internal constraints affect the scale of VLBI long-term solutions. The influence of the intra-technique combination model on the VLBI long-term solution is not significant. The influence of internal constraints on the long-term solution of SLR and that of the inter-technique combination model on the datum needs to be further studied.

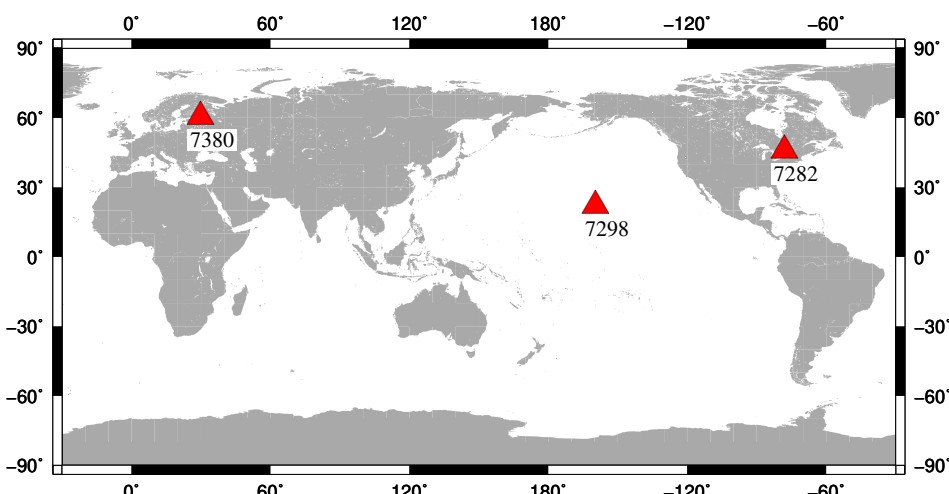

**Figure 6.** Station distribution of a session of less than five stations. The red triangle represents the VLBI station and its station code in the white box.

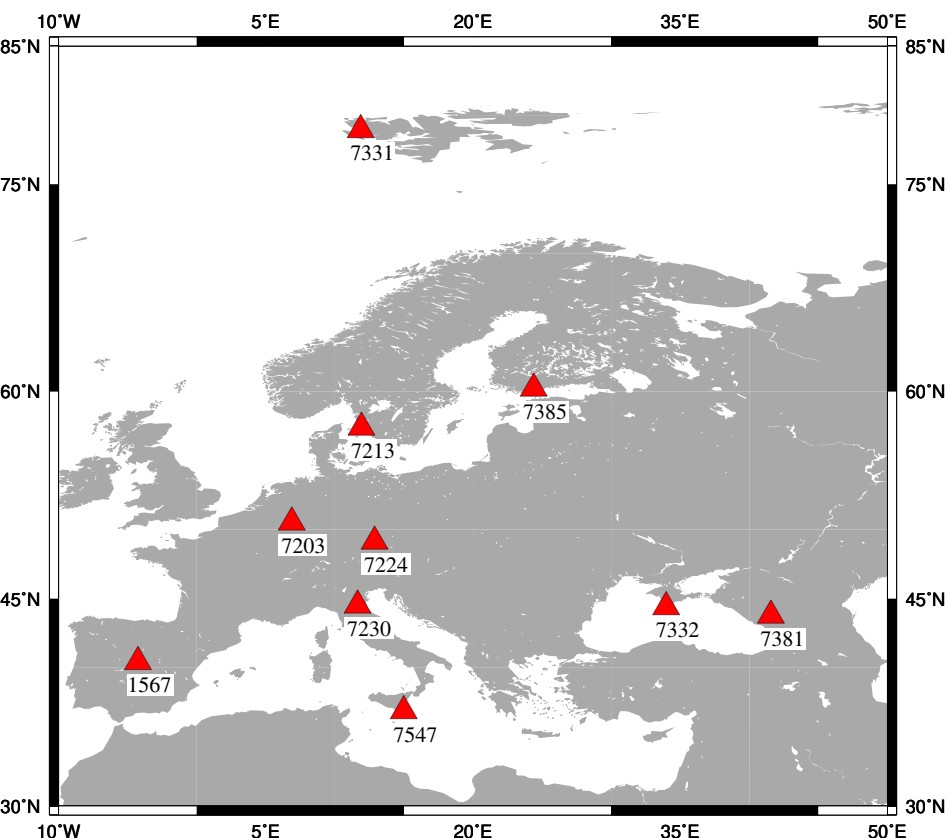

**Figure 7.** Station distribution of a session with regional coverage. The red triangle represents the VLBI station and its station code in the white box.

## 5. Conclusions

Different groups of scientists performed extensive serious research on constraints, against which our work is benchmarked. Inner constraints include internal constraints and kinematic constraints, and kinematic constraints are equivalent to minimum constraints. There is a strong correlation between the scale and translation parameters introduced by the VLBI intra-technique combination at IGN. Therefore, in order to satisfy the conditions of internal constraints, the scale and translation parameters absorb each other, thus interfering with the scale of VLBI. However, when the minimum constraints are applied, the VLBI long-term solution derived from the intra-technique combination model at IGN is equivalent to

that of DGFI-TUM. That is, the intra-technique combination model does not affect the scale of VLBI.

Different from internal constraints imposed on the transformation parameters, minimum constraints act on the station coordinates, complete the rank deficiency of the singular NEQs, and ensure that the accumulated solution is expressed in the identical TRF with the reference solution. Therefore, minimum constraints do not interfere with the inherent datum of space geodetic technology. Compared with the minimum constraint, there is a linear trend in the scale of VLBI long-term solutions realized by the internal constraint. According to the comparison between ITRF2014 and DTRF2014 (the time span of VLBI input data is between 1980.0 and 2015.0), this linear trend leads to a maximum offset of more than 4 mm. However, compared to DTRF2014, ITRF SLR has a negative scale rate (also leading to a maximum offset of more than 4 mm), which is the opposite of the VLBI rate. Therefore, the maximum difference between the VLBI and SLR scales is intensified. In future work, we will analyze the influence of the internal constraints on the SLR datum information and investigate whether the inter-technique combination model can affect the scale datum of VLBI and SLR.

**Author Contributions:** Conceptualization, S.S., G.W. and Z.Z.; methodology, S.S., G.W. and Z.Z.; software, S.S.; validation, S.S., G.W. and Z.Z.; formal analysis, S.S., G.W. and Z.Z.; resources, S.S., G.W. and Z.Z.; data curation, S.S., G.W. and Z.Z.; writing—original draft preparation, S.S., G.W. and Z.Z.; writing—review and editing, S.S., G.W. and Z.Z.; visualization, S.S., G.W. and Z.Z.; supervision, S.S., G.W. and Z.Z.; project administration, S.S., G.W. and Z.Z.; funding acquisition, G.W. and Z.Z. All authors have read and agreed to the published version of the manuscript.

**Funding:** This research was funded by the National Natural Science Foundation of China (CN) grant number 11873077 and the Natural Science Foundation of Henan Province of China.

**Institutional Review Board Statement:** Not applicable.

**Informed Consent Statement:** Not applicable.

**Data Availability Statement:** Publicly available datasets were analyzed in this study. These data can be found here: [https://cddis.nasa.gov/ (accessed on 22 January 2022)].

**Acknowledgments:** I'd like to thank Tianhe Xu from Shandong University, Weihai, and Naifeng Fu from School of Marine Science and Technology, Tianjin University for their help to this work. The authors would like to thank IVS for providing high-quality VLBI data.

**Conflicts of Interest:** The authors declare no conflict of interest.

## Abbreviations

The following abbreviations are used in this manuscript:

| | |
|---|---|
| TRF | Terrestrial Reference Frame |
| TRS | Terrestrial Reference System |
| VLBI | Very Long Baseline Interferometry |
| IGN | Institut National de l'Information Géographique et Forestière |
| ITRS | The International Terrestrial Reference System |
| SI | Le Système International d'Unités |
| TCG | Geocentric Coordinate Time |
| TT | terrestrial time |
| IAU | The International Astronomical Union |
| IUGG | The International Union of Geodesy and Geophysics |
| BIH | Bureau International de l'Heure |
| NNR | no net rotation |
| ITRF | The International Terrestrial Reference Frame |
| IERS | The International Earth Rotation and Reference Systems Service |
| GNSS | Global Navigation Satellite Systems |
| SLR | Satellite Laser Ranging |
| DORIS | Doppler Orbitography and Radio-positioning Integrated by Satellite |

| EOP | earth orientation parameter |
| --- | --- |
| NEQ | the normal equation |
| CC | combination center |
| DGFI-TUM | Deutsches Geodätisches Forschungsinstitut |
| JPL | Jet Propulsion Laboratory |
| DTRF2014 | DGFI-TUM's ITRS realization 2014 |
| PSD | postseismic deformation |
| SINEX | Solution Independent Exchange |
| ACs | the Analysis Centres |
| NNT | no-net-translation |
| MJD | Modified Julian Day |

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
