# Peer review of "Toward an Optimal Selection of Constraints for Terrestrial Reference Frame (TRF)"

_remotesensing, doi:10.3390/rs14051173_

Round 1

Reviewer 1 Report

This manuscript is clear, well organized and well written. A few editorial remarks are included in the annotated manuscript, but they are few. The manuscript is largely acceptable as is but this reviewer has a couple of comments.

  1. Item 2 in the list on the first page is correct, in that the definition of the ITRS is based on the SI meter. However, the authors may or may not be aware that in practice, the unit of length is not actually the SI meter. All lengths (and the gravitational constant GM) are reduced by 0.7 ppb due to the use of Terrestrial Time (TT) rather than TCG in the dynamical equations of motion. VLBI adds a small correction to be aligned with the SLR scale. (see, for example, page 172 of the 1991 Chapman Conference, or page 19 in the the IERS 2010 Conventions). This has no geodetic impact as long as all techniques work in the same system. I don’t know if this is necessary to bring up in the paper, though a footnote might be useful. I leave that choice to the authors

  1. This reviewer did not find figures 3 or 4 compelling or helpful in illustrating the level of correlations. Figure 3 is especially hard to interpret, since the ‘non-orthagonality’ is only a few tenths of a degree. Figure 4 is a little better, but I think both figures could be abandoned since Figure 5 is far more intuitive and illustrative, like Figure 7.

3) I would be inclined to disagree with the authors that there is a strong correlation between scale and the translation parameters based on any of the figures. Even in Figure 5, many of the correlations are quite small, and the high correlation days probably reflect poor geometry. It seems that the ‘average’ correlation is not at all bad for Tx, and not too bad for Ty and Tz. It is my experience that correlations approaching 0.9, while bothersome, do not seem to affect the reliability of the parameters I estimate. However, Figure 6 makes a good case that something is going on, so perhaps this level of correlation is indeed bothersome in this particular case.

Overall, the paper seems ready as is, though I would suggest the authors strongly consider recommendation 2 above. I think it would make the paper better.

Reviewer 2 Report

Please consider attached document

Author Response

This manuscript is a resubmission of an earlier submission. The following is a list of the peer review reports and author responses from that submission.